# Barriers to Radiotherapy Access in Sub-Saharan Africa for Patients with Cancer: A Systematic Review

**DOI:** 10.3390/ijerph21121597

**Published:** 2024-11-30

**Authors:** Portia N. Ramashia, Pauline B. Nkosi, Thokozani P. Mbonane

**Affiliations:** 1Department of Environmental Health, Faculty of Health Sciences, University of Johannesburg, Johannesburg 2000, South Africa; tmbonane@uj.ac.za; 2Faculty of Health Sciences, Durban University of Technology, Durban 4000, South Africa; paulinen1@dut.ac.za

**Keywords:** access barriers, infrastructure, disparities, socioeconomic status, geographic location, factors, health systems, cultural beliefs, cancer, radiotherapy, systematic review

## Abstract

Background: Access to radiotherapy services is critical for effective cancer treatment, yet patients in sub-Saharan Africa face numerous barriers to accessing these services. The region is experiencing a significant increase in cancer cases, with a more than 85% increase in cancer cases reported in the past decade, highlighting the critical role of radiotherapy in enhancing patient prognosis. This systematic review aims to explore the barriers to radiotherapy access in sub-Saharan Africa. The barriers explored will be used to inform the development of the framework to improve access to radiotherapy in the Gauteng provinces, South Africa. Methods: A systematic search of electronic databases was conducted to identify relevant studies published between January 2013 and December 2023. Studies reporting on barriers to radiotherapy access in SSA were included and put into four categories of barriers: health system factors, patient sociodemographic factors, patient factors, and provider factors. Data were synthesised using thematic analysis. Results: This review identifies geographical, financial, cultural, logistical, and systemic barriers to radiotherapy access in sub-Saharan Africa, including limited infrastructure, long travel distances, and inequitable distribution of services. Systemic barriers, including policy gaps and governance issues, also contribute to the inequitable distribution of radiotherapy services in the region. Conclusions: This systematic review highlights the diverse array of barriers to radiotherapy access in sub-Saharan Africa and emphasises the urgent need for targeted interventions to address these challenges.

## 1. Introduction

Cancer is a growing public health concern in sub-Saharan Africa (SSA), with an estimated 1.06 million new cancer cases in 2020. The incidence of cancer is projected to increase significantly in the coming years. Despite this, access to radiotherapy services, a crucial component of cancer treatment, remains severely limited in SSA [1,2]. Radiotherapy plays an important role in the management of cancer, particularly in cases where surgery or chemotherapy alone is not sufficient. It involves using high-energy radiation to destroy cancer cells or stop their growth [3,4]. However, access to radiotherapy services in SSA is severely lacking [5]. According to a 2013 report, most countries in Africa lacked access to radiotherapy as part of their overall cancer treatment strategies [1]. This gap has persisted, with a continued shortage of teletherapy units and brachytherapy afterloaders across the continent [6]. In 2013, it was estimated that there was a total of 277 radiotherapy machines serving a population of 1 billion individuals in Africa. This translates to approximately one machine per 3.6 million people, which is far below the International Atomic Energy Agency recommendation of one machine per 250,000 people. The disparity is even more pronounced when comparing high-income countries, where there is one radiotherapy machine for every 120,000 people, to middle-income countries, where one machine serves over 1 million people, and low-income countries in Africa, where approximately 5 million or more people on average rely on a single radiotherapy machine [1,2,6].

The population of interest in this systematic review is cancer patients in SSA. This region of Africa lies south of the Sahara Desert, encompassing countries such as Nigeria, South Africa, Kenya, and Tanzania. The problem or phenomenon of interest is the limited access to radiotherapy for cancer treatment in SSA. The contextualisation of this issue is crucial to understand the gravity of the problem. SSA faces multiple challenges when it comes to cancer care, including limited healthcare facilities, inadequate infrastructure, a shortage of trained healthcare professionals, and insufficient resources for cancer treatment. These factors combine to create a scenario where a large proportion of cancer patients in the region do not have access to radiotherapy services [7,8]. This systematic review aims to answer the following question: ‘What are the barriers to radiotherapy access for cancer patients in SSA, as identified in the literature?’ This review draws upon health access frameworks and implementation science literature to systematically analyse the barriers hindering radiotherapy access in SSA, aiming to inform policy interventions and improve cancer care delivery. While previous studies have examined access to treatment in SSA, there is a need for a comprehensive, updated systematic review synthesising the evidence on barriers specific to radiotherapy. This review addresses this gap by including different adult cancers from various countries in SSA. By analysing the barriers to radiotherapy access in different countries in SSA, this review aims to identify common barriers to radiotherapy access for cancer patients in SSA.

### The Significance of This Systematic Review

For several reasons, understanding the current state of radiotherapy access in SSA is important. Firstly, it helps policymakers and healthcare organisations identify investment and resource allocation priority areas. By knowing where the deficiencies lie, targeted interventions and improvements can be made to strengthen the healthcare infrastructure and enhance access to radiotherapy. Secondly, this review would contribute to the existing body of evidence on cancer care in SSA. It would provide valuable insights into the specific challenges faced by cancer patients in accessing radiotherapy services, which can inform the development of tailored interventions and strategies. Additionally, the findings of this review can help establish benchmarks and indicators to track progress over time and compare the performance of different countries or regions.

This systematic review explores the barriers to radiotherapy access in SSA. While understanding enablers is valuable, this review focuses on barriers because identifying and analysing barriers is crucial for developing targeted interventions and policies to improve service delivery. This approach aligns with the goal of the postgraduate study that is aimed at developing a framework to improve radiotherapy access in Gauteng Province, which requires a clear understanding of the obstacles to address. The findings of this review will be used in conjunction with other quantitative and qualitative results to inform the development of the framework to improve access to radiotherapy in Gauteng province, South Africa.

## 2. Materials and Methods

To comprehensively understand the barriers to radiotherapy access in SSA, this study adopted a systematic review approach. Systematic reviews offer a rigorous and comprehensive method for synthesising existing evidence on a specific topic. This approach allows for the identification of patterns, trends, and gaps in the literature, which is crucial for understanding complex issues like the multifaceted challenges hindering radiotherapy access in SSA. This review is conducted for the purpose of providing insights for future phases of the current postgraduate research study that aims to develop a framework aimed at improving access to radiation therapy in Gauteng province. This systematic review will be conducted in accordance with the Preferred Reporting Items for Systematic Reviews and Meta-Analyses guidelines. This review was registered with the PROSPERO International Prospective Register of Systematic Reviews (Registration number: CRD42024500362)

### 2.1. Inclusion and Exclusion Criteria

Articles were eligible for inclusion in this systematic review of studies if they were conducted in SSA, report on barriers to accessing radiotherapy services, including but not limited to geographic accessibility, financial barriers, socioeconomic factors, cultural beliefs and attitudes, infrastructure and resources, and policy and governance issues, and were published between January 2013 and December 2023 in English. This timeframe was selected because the year 2013 marked a period of increased global attention to cancer control in low- and middle-income countries, potentially leading to a higher volume of relevant publications [9]. The search was conducted between February 2024 and July 2024. No prior restrictions existed concerning the study design, whether qualitative, quantitative, or mixed methods. Studies conducted outside of SSA, do not specifically address barriers to accessing radiotherapy, and are without abstracts or for which the full text was not available were excluded. No studies were excluded after the quality assessment.

### 2.2. Study Selection Strategy

This systematic review was conducted according to the Preferred Reporting Items for Systematic Reviews and Meta-Analyses (PRISMA) statement by [10]. A comprehensive search strategy was developed, and an exhaustive search for studies was conducted in different databases: Medline (PubMed), ScienceDirect, African Journals Online, Mendeley, ResearchGate, and Google Scholar. While we acknowledge the value of the grey literature in providing potentially valuable insights, due to time and resource limitations, we were unable to include it in the present review. Additionally, our review focused primarily on peer-reviewed publications to ensure a certain level of methodological rigour and minimise potential bias. However, we recognise that future research could benefit from exploring the grey literature on this topic to provide a more comprehensive understanding of barriers to radiotherapy access in SSA.

To make the search exhaustive and identify additional articles, we looked for other sources and carried out country-by-country (48 sub-Saharan African countries) searches. Reference lists of relevant articles were also hand-searched. The following keywords were combined by Boolean operators “AND”, “OR”, and “Not” to obtain several search equations according to the databases: “Breast cancer”; “Breast carcinoma”; “Breast neoplasm”; “Breast Tumor”; “Factors”; “Determinants”; “Barriers”; “Challenges”; “Delayed treatment”; “Time-to-Treatment”; “Provider delay”; radiotherapy delay”; “Treatment delay”; “Health system delay”; “Healthcare delivery”; “healthcare access”; “health service accessibility”; “Africa”; “sub-Saharan Africa”; “low-middle income”; and the names of each of the 48 sub-Saharan African countries.

The PICO (Population, Intervention, Comparison, and Outcome) framework was used as a guide to developing the systematic review question, “What factors contribute to barriers to radiotherapy access in SSA?” The first stage involved the use of keywords developed based on the PICO framework; a total of 1203 articles were discovered in the first stage (keyword screening), 974 in the second stage (title and abstract), and 338 were found to be relevant. The third stage was the full text screening, where full texts of all potentially eligible papers were retrieved and reviewed for inclusion in this review according to the inclusion criteria. All included studies were independently reviewed by two authors to confirm eligibility, and 91 met the criteria for this paper’s results review (Figure 1).

### 2.3. Data Extraction and Items

For the included studies, two authors (PNR, TPM, or PBN) independently extracted information such as the characteristics of the study (title, authors, year of publication, country, study design, research method, participants, and sample size), barrier categories, reported barriers, and the study conclusion. All barriers were classified as follows: 

Health systems: this category encompasses barriers related to the organisation, accessibility, and affordability of healthcare services, including factors such as inadequate infrastructure, limited availability of radiotherapy facilities, long waiting times, and high treatment costs;

Sociodemographic factors: this category includes barriers related to patients’ social and economic circumstances, such as marital status, low income, lack of education, rural residence, and cultural beliefs that may hinder access to radiotherapy;

Patient preferences: this category focuses on barriers arising from patients’ personal choices or circumstances, such as fear of treatment side effects, mistrust in the healthcare system, or competing priorities that may lead to delayed or refused treatment;

Provider factors: this category encompasses barriers related to healthcare providers’ knowledge, attitudes, and practices, including factors such as limited awareness of radiotherapy options, referral delays, or biases that may influence treatment decisions. 

Any discrepancies in the process of selection and extraction were resolved through discussion. In this review, we focused on the barriers to accessing radiation therapy, but we also included studies that focus on the effect of late presentation for diagnosis, which affects the delayed start of treatment [10].

### 2.4. Quality Assessment

The quality of the qualitative studies was assessed by using the Critical Appraisal Skills Program (CASP) quality-assessment tool [11]. The quantitative studies were evaluated utilising the National Institute of Health (NIH) Quality Assessment Tool for Observational cohort and cross-sectional studies. The quality of the study was evaluated based on several criteria: research question, study population, eligibility criteria, justification of sample size, outcome measures, response and follow-up rates, statistical analyses, and ethical considerations. Furthermore, confidence in the evidence from the Reviews of Qualitative Research (CERQual) tool [12] was employed to evaluate the evidence for each qualitative finding. Qualitative findings were categorised into three levels of confidence—high, moderate, and low—based on an evaluation of four components: methodological limitations, relevance, adequacy, and coherence [12].

## 3. Results

### 3.1. Study Characteristics

Table 1 summarises the primary characteristics of the included studies. Of the 91 research articles incorporated in this review, 63 were quantitative, 12 were qualitative, 4 employed mixed methods (quantitative and qualitative approaches), and 12 were reviews. These studies were conducted across 13 countries in sub-Saharan Africa (SSA). Specifically, 17 (19%) studies were conducted in East Africa, 18 (20%) in West Africa, 22 (24%) in Southern Africa, and 34 (37%) were multi-country studies encompassing countries from all three regions of SSA (East, West, and Southern Africa). The publication dates of the studies ranged from 2013 to 2023. The sample sizes for the quantitative and mixed-method studies varied from 23 to 4863 participants. The qualitative articles utilised focus group discussions and in-depth interviews to explore barriers to radiotherapy access for cancer patients.

The five most frequently identified themes were the following: treatment centre characteristics (n = 58), socioeconomic status (n = 48), distance to the treatment centre (n = 44), cultural beliefs (n = 27), and waiting times (n = 21). Among these, three themes were categorised as barriers within health system factors. Under the category of patient preferences, four themes were identified, including cultural beliefs: patient preference (n = 17), age (n = 7), and life expectancy (n = 1). Within the provider factors category, two themes emerged: lack of understanding/awareness (n = 16) and lack of referral (n = 7). A total of 65 studies (71.4%) addressed more than one theme, while 7 articles (7.8%) addressed all four categories, resulting in some articles being classified under multiple headings. In certain studies, the theme was not the primary focus of the investigation and may have been addressed only briefly; however, it was still deemed relevant by the reviewers.

### 3.2. Thematic Analysis

#### 3.2.1. Health System Factors

In sub-Saharan Africa (SSA), the characteristics of treatment centres, proximity to these facilities, and waiting times constitute inter-related dimensions of the health system barriers that impede access to radiation therapy [13]. Collectively, these factors contribute to significant challenges in obtaining timely and high-quality radiation therapy services, underscoring the necessity for comprehensive strategies aimed at enhancing healthcare infrastructure, improving geographical accessibility, and minimising waiting times for patients requiring radiation therapy.

The health system comprises a variety of interconnected components, including healthcare infrastructure, workforce capacity, financing mechanisms, referral systems, and quality assurance measures [14]. Within SSA, numerous challenges inherent in the health system serve as obstacles to accessing radiation therapy. Radiation therapy centres are predominantly situated in urban locations, often within capital cities, thereby creating barriers for rural populations seeking to utilise these services. The extensive distances and inadequate transportation infrastructure prevalent in many regions of SSA further exacerbate this issue [13,15,16]. Additionally, the limited infrastructure is compounded by challenges related to reliable electricity and water supply. Without consistent sources of power, the operation of advanced radiation therapy equipment becomes problematic, resulting in disruptions to treatment schedules. Furthermore, there exists a deficiency in trained radiation oncologists, medical physicists, and radiation therapists within SSA [17]. The articles reviewed in this study identified three primary themes under the category of health system barriers.

##### Characteristics of Treatment Centres

Limited healthcare infrastructure and resources contribute to the challenges faced by treatment centres in SSA. Many centres lack essential equipment, trained personnel, and facilities for delivering radiation therapy services effectively [5,17,18,19,20,21,22,23,24]. This limitation in treatment centre characteristics directly impacts the quality and availability of radiation therapy services. Patients may face longer waiting times due to the limited capacity of treatment centres to accommodate the demand for services [25]. In SSA, there are 430 radiation therapy machines in twenty-six countries distributed in a total of 120 RT centres; South Africa comprises 51% of these centres, highlighting disparities between the Republic of South Africa and other SSA nations [26]. Despite a significant increase in radiation therapy capacity, there remains a persistent issue with both temporary and permanent breakdowns of existing machines, resulting in insufficient provision of quality radiation therapy services [27,28]. The shortage of health workers was also identified as a factor delaying patients’ access to treatment; this scarcity limits the capacity of existing treatment centres and impedes efforts to establish new ones. Additionally, retention of skilled personnel is difficult due to better opportunities and higher salaries abroad [18,19,28,29].

##### Distance to Treatment Centres

According to studies [15,19,30,31], in SSA, treatment centres offering radiation therapy services are often concentrated in urban areas, making access difficult for patients residing in rural or remote regions. This geographical disparity in access exacerbates the challenge of distance to treatment centres. Patients living far from treatment centres may experience prolonged travel times and increased transportation costs when seeking radiation therapy [32,33,34,35,36]. These factors lead to delays in treatment access and may discourage patients from pursuing care entirely. Several studies indicate that the significant distances patients in SSA must travel to receive radiotherapy treatment are attributable to the limited availability of treatment centres in the region. The travel distances for patients in Nigeria and South Africa are shorter than those for patients in Tanzania, likely due to the higher number of radiotherapy centres in the former two countries. Nonetheless, the reduced travel distances for patients in Nigeria and South Africa do not fully reflect the overall travel burden linked to radiotherapy treatment, as traffic congestion in major SSA cities, such as Lagos, can significantly increase travel times for patients [37].

In many parts of SSA, transportation infrastructure is underdeveloped, with limited road networks and public transportation options. Patients may face difficulties in accessing treatment centres due to a lack of reliable transportation, especially in remote areas. Furthermore, travelling long distances to reach a radiation therapy centre comes with a significant financial burden on patients and their families. Costs may include transportation fees, accommodation, food, and loss of income due to time away from work. For most patients in the region, these expenses are prohibitive, leading to delays or avoidance of seeking treatment [38].

##### Waiting Times

This review indicates that treatment delays in women with breast cancer in SSA are affected by various interconnected health system factors. Extended waiting times for radiotherapy are frequently observed in SSA [9,39,40,41]. Some facilities report waiting times exceeding three months. Waiting time recommendations range from 4 to 8 weeks. Extended waiting times exceeding three months lead to unfavourable outcomes [42]. Prolonged waiting times for radiotherapy, along with delays in other treatment pathways observed in various low- and middle-income countries in SSA, adversely impact survival rates [17,32,43].

Limited treatment centre capacity and high demand for radiation therapy services result in extended waiting times for patients in SSA. Patients may face delays in scheduling appointments, receiving treatment planning, and starting radiation therapy sessions. Longer waiting times are often exacerbated by factors such as equipment breakdowns, staff shortages, and bureaucratic processes within treatment centres [39,44]. Patients may experience frustration and anxiety while waiting for treatment, which can impact their overall care experience and treatment outcome. Furthermore, this barrier in many studies has been linked to other barriers, including the demand for radiation therapy services often exceeding the available supply in many SSA countries. This high demand may be due to a variety of factors, including the high prevalence of cancer, late-stage diagnoses, and limited alternative treatment options; geographical disparities also exist, where patients living in rural or remote areas may face longer waiting times compared to those residing in urban areas with better access to radiation therapy centres. The limited availability of treatment facilities in rural areas and challenges in transportation exacerbate this disparity [25,32,33].

#### 3.2.2. Sociodemographic Factors and Patient Factors

The inter-relation of socioeconomic status, cultural beliefs, patients’ preferences, age, and life expectancy underscores the complex sociodemographic and patient factors influencing access to radiation therapy in SSA [45,46]. The articles in this review reported the following themes under the sociodemographic factors and patient factors barrier categories:

##### Socioeconomic Status and Patient Preferences

Studies have shown that demographic factors, such as marital status and employment status, are associated with experienced barriers [47,48,49,50]. Lower socioeconomic status directly impacts access to radiation therapy in SSA. Individuals with limited financial resources may struggle to afford transportation costs, out-of-pocket expenses, and time off work for treatment sessions. Socioeconomic status also influences access to health insurance coverage. Individuals without insurance or with limited coverage may face significant financial barriers to accessing radiation therapy, further exacerbating disparities in care. Married patients reported a greater number of barriers compared to those who were never married or divorced. Married women may experience increased familial responsibilities, reduced disposable income due to larger family size, or a lack of agency in household decision-making dominated by husbands [34]. Employment status, alongside marital status, significantly influenced the number of reported barriers. Most study participants were unemployed, potentially indicating that they were homemakers primarily responsible for family care. Employed patients exhibited a significantly lower likelihood of reporting financial barriers compared to their counterparts [39,51,52,53]. Cultural beliefs play a significant role in shaping perceptions of illness and treatment preferences in SSA [54]. Some cultural beliefs may stigmatise cancer and its treatment, leading to reluctance or refusal to seek radiation therapy. Cultural beliefs may also influence preferences for traditional healing practices over biomedical treatments like radiation therapy [51,55,56]. Individuals may prioritise traditional healers or spiritual remedies based on cultural familiarity and trust. Patients’ preferences influence their decision-making regarding radiation therapy. Factors such as fear of treatment side effects, mistrust of biomedical treatments, and preference for alternative therapies may lead patients to decline or delay radiation therapy [57]. Seven studies described age as a barrier to accessing radiotherapy [37,51,58,59,60,61,62]. They outlined the extent to which age and literacy were associated with medical attention-seeking, with awareness of cancer treatment among 50–59 years at 38.7% compared to those between the ages of <30 and >60. Unlike in other regions, this barrier was not reported as a stand-alone barrier [20]. Life expectancy indirectly influences access to radiation therapy in SSA by affecting cancer incidence, treatment decisions, and resource allocation within healthcare systems. Patients with shorter life expectancies may prioritise palliative care over curative treatments like radiation therapy, particularly if they perceive treatment as burdensome or unlikely to significantly extend their lifespan [63].

#### 3.2.3. Provider Factors

A link between the health system and provider factors highlights the complex challenges individuals seeking access to radiation therapy in SSA face [64,65]. There are two themes in this barrier category: lack of understanding or awareness and lack of referral, hindering access to radiation therapy in SSA. These barriers result in inefficient referral pathways, lack of coordination between primary care providers and treatment centres, and delays in diagnosis, contributing to delays in accessing treatment [66].

##### Lack of Understanding or Awareness

Lack of understanding and awareness about radiation therapy has been reported by 16 studies as a barrier to accessing radiation therapy in SSA [13,17,22,25,32,34,35,39,60,67,68,69,70,71]. Many individuals in SSA, particularly in rural and underserved areas, have limited knowledge about cancer and its treatment options. This lack of awareness may result in delayed diagnosis, reluctance to seek medical care, and misconceptions about the effectiveness and safety of treatments like radiation therapy [71]. The stigma surrounding cancer and its treatment is prevalent in many SSA communities [22]. Misconceptions about radiation therapy, including fears about its side effects or beliefs that it can worsen the disease, may contribute to reluctance or refusal to undergo treatment. Fear of social stigma and discrimination may also deter individuals from seeking radiation therapy [20]. Studies by Akuoko et al. (2017) [21] and Omotoso et al. (2023) [64] mention that access to accurate and reliable information about cancer and its treatment is often limited in SSA, particularly in rural and remote areas where healthcare infrastructure is inadequate.

##### Lack of Referral

Lack of referral serves as a significant barrier to accessing radiation therapy in SSA was reported by seven studies [13,17,66,67,68,69,70,71]. Healthcare systems in SSA often lack robust referral systems and pathways for patients needing specialised cancer care, including radiation therapy [63,67]. Inadequate communication and coordination between primary care providers, referral hospitals, and specialised cancer centres can result in delays or missed opportunities for timely referrals [13,71]. Furthermore, healthcare providers, especially in rural and remote areas, may lack adequate training and education about cancer diagnosis and treatment, including when to refer patients for radiation therapy. Without proper knowledge and awareness, healthcare providers may miss opportunities to identify patients who could benefit from radiation therapy [56,68].

## 4. Discussion

This systematic review, the first to comprehensively explore barriers to radiotherapy access across all cancer types in SSA, revealed a complex interplay of systemic, financial, logistical, and sociocultural factors. Our findings corroborate the existing literature on specific cancer types, while also highlighting the broader context of access challenges across the cancer spectrum. The diverse range of barriers identified aligns with Penchansky and Thomas’s multidimensional model of access, emphasising the need for interventions targeting multiple dimensions to improve radiotherapy utilisation in SSA. Furthermore, the impact of financial barriers is exacerbated by logistical challenges, creating compounded disadvantages for patients in rural areas who face high transportation costs and limited access to treatment facilities. These interconnected barriers underscore the need for comprehensive strategies that address both individual-level and system-level challenges.

The studies included in this review revealed a complex interaction of barriers hindering access to radiotherapy services in SSA, broadly categorised into health system, sociodemographic, patient, and provider factors. Health system barriers, such as the severe shortage of radiotherapy equipment and limited infrastructure, particularly in rural areas, directly affect the availability of this essential service. The combined findings of Bishr and Zaghloul [72] (Bishr and Zaghloul, 2018) and Bhatia et al. [1] (Bhatia et al., 2019) paint a concerning picture of radiotherapy access in SSA. Bishr and Zaghloul highlight the limited availability of linear accelerators, and Bhatia et al. reveal critical shortages in human resources and other essential components of radiotherapy delivery. This convergence of infrastructure and workforce limitations creates a significant bottleneck in access, hindering efforts to provide timely and effective cancer care to patients in need and creating a significant barrier to access, as highlighted by Penchansky and Thomas’s framework. This shortage is often exacerbated by weak health systems and policy gaps, forcing patients to travel long distances for treatment [73,74]. Geographical barriers, including long travel distances and inadequate transportation infrastructure, severely impact accessibility to radiotherapy services. Moreover, the logistical challenges, as emphasised by Penchansky and Thomas, disproportionately affect individuals in remote areas.

While distance and travel time represent significant barriers to radiotherapy access in both developed countries and SSA, as highlighted in Thompson et al. [73], the underlying context differs significantly. In developed countries, the challenge often lies in the distribution of radiotherapy centres, with some rural areas lacking nearby facilities. However, the overall availability of advanced radiotherapy technology and trained personnel is generally higher. In contrast, SSA faces a more fundamental scarcity of resources, with limited access to both equipment and qualified professionals. This disparity in resource availability underscores the greater magnitude of the challenge in SSA.

These long journeys create a significant financial burden, and the high cost of radiotherapy, often requiring substantial out-of-pocket payments, poses a major affordability challenge for many patients. This financial burden, exacerbated by limited health insurance coverage, aligns with Penchansky and Thomas’s emphasis on the financial dimension of access with patients often selling assets or borrowing money to cover travel and accommodation costs, pushing them further into poverty. This financial strain is further compounded by lost income due to time away from work, impacting not only the patient but also their families. Donkor et al. [75] highlight the devastating impact of these costs, particularly for individuals from low-income backgrounds. Their study reveals that patients often deplete their savings, incur debt, or forgo treatment altogether due to financial hardship. This financial toxicity not only affects individual patients but also has broader societal implications, as families struggle to cope with the economic consequences of cancer care. Furthermore, the lack of robust social safety nets in many SSA countries exacerbates these financial challenges, leaving patients with limited support to navigate the economic burden of radiotherapy.

Moreover, the long waiting times, inflexible appointment scheduling, and limited-service hours create accommodation barriers, making it difficult for patients to access radiotherapy services in a timely and convenient manner. This dimension, highlighted by Penchansky and Thomas, highlights the importance of organising services to meet patient needs. This shortage is partly attributed to limited training opportunities within the region and brain drain to higher-income countries. These health system challenges disproportionately affect patients from low socioeconomic backgrounds who often lack the resources to overcome these barriers. 

Sociocultural beliefs and practices also play a significant role in influencing radiotherapy access. These beliefs and practices, as noted by Penchansky and Thomas, can lead to delays in seeking care and reduced adherence to treatment plans. Krah et al. [76] explored the role of traditional healers in rural northern Ghana, revealing that many individuals initially seek care from traditional healers before considering conventional medical treatments. This reliance on traditional healing systems can delay diagnosis and treatment, potentially leading to worse outcomes for cancer patients. Furthermore, the stigma associated with cancer diagnoses, as documented in Krah et al., can deter individuals from seeking timely care or disclosing their condition to family and community members. These sociocultural factors underscore the need for culturally sensitive approaches to cancer care that address traditional beliefs and promote open communication about cancer. Additionally, the study by Thompson et al. [73] highlighted the importance of carer support and seasonal weather as factors influencing access in developed countries. While these factors also play a role in SSA, the challenges are heightened by weaker health systems, limited infrastructure, and a shortage of trained healthcare professionals. This disparity in health system capacity exacerbates the impact of these practical barriers.

The conceptualisation of the four barrier categories using the Penchansky and Thomas framework highlights the complex challenges individuals seeking access to radiation therapy in SSA face. Addressing these barriers requires comprehensive strategies to strengthen healthcare infrastructure, increase workforce capacity, mobilise financial resources, improve referral systems, enhance quality assurance measures, and strengthen health information systems.

### Strengths and Limitations of This Review

This systematic review focused on barriers to radiotherapy access for all cancers in SSA, not just breast or cervical. The search strategy was refined to ensure comprehensiveness, but some relevant articles may not have been collected. The scope of this review was limited to the recent SSA literature, potentially excluding important work. The lack of a strong cutoff point for determining the prevalence of an issue in each article led to some barriers being the primary focus of some studies but only mentioned in passing in others. Despite these limitations, this review identified the general scope of barriers to access and highlighted significant trends. This is the first systematic review to focus on barriers to radiotherapy access for all cancers in SSA. Furthermore, this review provided valuable insights for future phases of the current study that aim to improve access to radiation therapy in Gauteng province to address these barriers.

## 5. Conclusions

This review systematically analysed barriers to radiotherapy access in SSA, drawing upon health access frameworks and the implementation science literature. Our findings reveal a complex interplay of financial, geographical, and sociocultural barriers, compounded by limited infrastructure and workforce capacity. These findings underscore the urgent need for multi-sectoral interventions. Policy recommendations include increased government funding for radiotherapy infrastructure and the implementation of financial assistance programs for patients. This review serves as a critical foundation for the postgraduate study, which will develop a framework to improve radiotherapy access in Gauteng, South Africa, translating these findings into targeted interventions to improve cancer care delivery in the region. By addressing these barriers, we can move towards equitable access to life-saving radiotherapy services and improve cancer outcomes across SSA.

## Figures and Tables

**Figure 1 ijerph-21-01597-f001:**
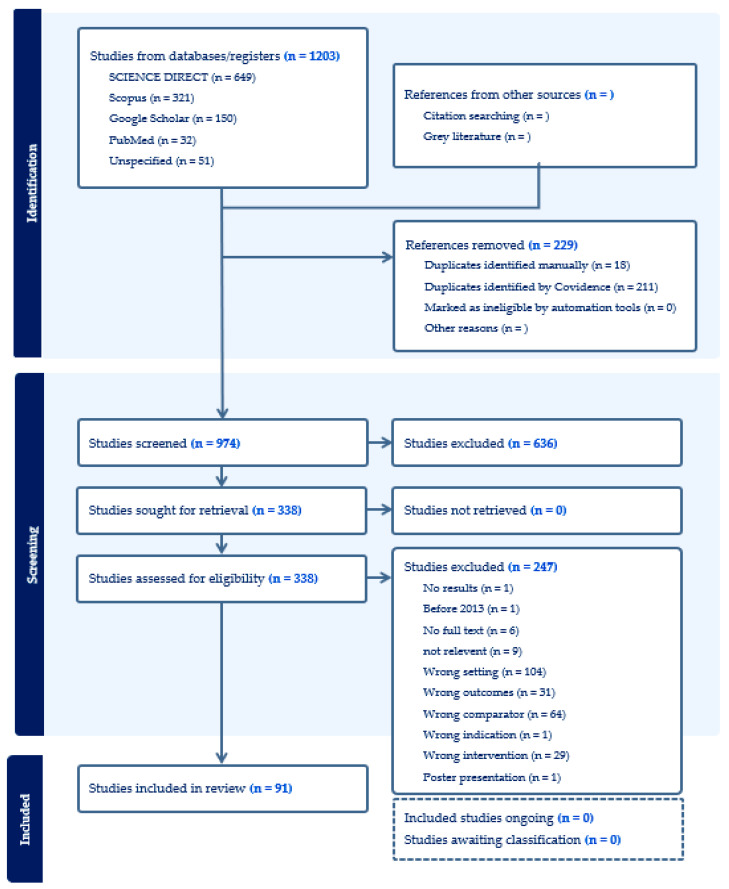
Preferred Reporting Items for Systematic Reviews and Meta-Analyses (PRISMA) flow diagram.

**Table 1 ijerph-21-01597-t001:** Main characteristics of the included studies.

Category	Reported Barrier	Percentage (%)	Reference *
Health system	Treatment Centre Characteristics	58	2,3,4,5,6,8,9,10,11,12,15,16,17,18,19,20,21,23,26,27,29,32,35,36,377,38,39,40,4,45,47,49,50,51,52,53,54,56,57,58,63,64,66,67,68,69,70,74,75,77,78,81,83,87,89,90,91
	Wait times	21	5,11,12,14,23,33,35,36,37,38,47,56,66,77,78,84,86,87,88,89
	Distance to treatment centre	44	7,11,12,13,21,24,25,27,28,31,33,35,37,38,42,47,48,50,52,53,54,55,56,59,60,62,63,64,65,66,70,71,73,75,77,78,79,80,83,84,85,88,90,91
Sociodemographic factors	Socioeconomic status	48	1,5,6,7,12,22,26,29,30,31,34,35,36,38,41,42,43,44,45,46,47,48,50,52,55,56,57,58,60,61,62,66,67,68,71,72,75,76,78,80,82,83,87,88,89,90
Patient factors	Age	7	7,22,41,42,43,72,90
	Cultural beliefs	27	6,7,12,14,21,26,29,30,35,36,38,41,42,45,46,47,50,52,53,56,58,59,61,63,78,88,89
	Patient preference	17	12,14,21,23,26,29,30,31,35,6,47,52,54,56,58,61,64,78
	Life expectancy	1	1
Service Provider factors	Lack of referral	7	16,26,33,47,53,75,81
	Lack of understanding/awareness	16	2,6,33,42,46,47,53,56,58,63,64,71,75,83,88,89

* Numbering as per the data extraction table (Appendix A).

## Data Availability

All data in this study were provided in the main manuscript.

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
