# Peer review of "Barriers to Radiotherapy Access in Sub-Saharan Africa for Patients with Cancer: A Systematic Review"

_ijerph, 2024, doi:10.3390/ijerph21121597_

Round 1

Reviewer 1 Report

Comments and Suggestions for Authors

In the attached file we could find my revision.

Comments on the Quality of English Language

None

Reviewer 2 Report

Comments and Suggestions for Authors

Overall, the paper is very well written scientifically.  The study's categorization of barriers—systemic, demographic, logistical, and provider-related—offers valuable insights. However, the review would benefit from a deeper analysis of how specific interventions might mitigate these challenges and a stronger focus on region-specific variations within sub-Saharan Africa. I also emphasized to add some relevent studies. 

Reviewer 3 Report

Comments and Suggestions for Authors

This manuscript provides a comprehensive review of barriers to accessing radiotherapy in Sub-Saharan Africa (SSA), identifying key challenges related to healthcare infrastructure, sociocultural factors, and healthcare provider constraints. The main findings emphasize significant barriers, including limited treatment centers, equipment shortages, extended waiting times, financial constraints, cultural stigma, and inadequate referral systems. The review underscores the need for strengthened healthcare infrastructure, improved referral systems, capacity-building initiatives, and policy reforms to address these issues. It is recommended that the authors provide more specific examples of regional collaborations and innovative approaches to address equipment shortages and treatment delays. Additionally, a more in-depth exploration of policy interventions, such as expanding insurance coverage or implementing transportation subsidies, would enhance the discussion on alleviating financial barriers. Expanding on how cultural beliefs impact treatment access and exploring the integration of traditional healing practices with modern medicine could further enrich the paper. Including insights into the potential of digital health solutions, such as telemedicine, to improve referral systems would also add value. Furthermore, future research could focus on evaluating the effectiveness of existing interventions or pilot programs designed to improve radiotherapy access in SSA.

Round 2

Reviewer 1 Report

Comments and Suggestions for Authors

none